# The Golden Egg: Nutritional Value, Bioactivities, and Emerging Benefits for Human Health

**DOI:** 10.3390/nu11030684

**Published:** 2019-03-22

**Authors:** Sophie Réhault-Godbert, Nicolas Guyot, Yves Nys

**Affiliations:** Biologie des Oiseaux et Aviculture, INRA, Université de Tours, 37380 Nouzilly, France; nicolas.guyot@inra.fr (N.G.); yves.nys@inra.fr (Y.N.)

**Keywords:** chicken egg, food, nutrients, bioactivity, variability, digestion, health

## Abstract

Egg is an encapsulated source of macro and micronutrients that meet all requirements to support embryonic development until hatching. The perfect balance and diversity in its nutrients along with its high digestibility and its affordable price has put the egg in the spotlight as a basic food for humans. However, egg still has to face many years of nutritionist recommendations aiming at restricting egg consumption to limit cardiovascular diseases incidence. Most experimental, clinical, and epidemiologic studies concluded that there was no evidence of a correlation between dietary cholesterol brought by eggs and an increase in plasma total-cholesterol. Egg remains a food product of high nutritional quality for adults including elderly people and children and is extensively consumed worldwide. In parallel, there is compelling evidence that egg also contains many and still-unexplored bioactive compounds, which may be of high interest in preventing/curing diseases. This review will give an overview of (1) the main nutritional characteristics of chicken egg, (2) emerging data related to egg bioactive compounds, and (3) some factors affecting egg composition including a comparison of nutritional value between eggs from various domestic species.

## 1. Introduction

In 1968, the egg industry had to face American Heart Association recommendations that encouraged people to consume less than three whole eggs per week claiming that high dietary cholesterol equals high blood cholesterol and consequently higher cardiovascular disease risks. These recommendations impacted, not only the egg industry, but also partly influenced people’s dietary habits depriving them from an affordable food of high nutritional interest. By 1995, there was a concerted effort to unify all the US national dietary recommendations and to support in vitro and in vivo research to rehabilitate eggs [1]. Half a century of research has now demonstrated that egg intake is not associated with increased health risk [2] and that it is worth incorporating such product in our diet with regard to its high nutrients content and its numerous bioactivities [1]. Some recent researches have highlighted the beneficial role of eggs for humans, including physically active people, and several authors have demonstrated that egg cholesterol was not well absorbed [3,4]. Consequently, consuming eggs does not significantly impact blood cholesterol concentration [3,4]. In parallel, egg consumers especially 6–24 month-old infants eat lower added and total sugars relative to non-consumers [5], which is likely correlated with its satiety effect [2,6,7]. It is now well established that egg can contribute to overall health across the life span, although people suffering from metabolic disorders such as diabetes, hypercholesterolemia, and hypertension still need to take caution with their dietary cholesterol intake [8]. Another concern relates to egg allergy, which is a common infant food allergy with a prevalence estimated to be between 1.8% and 2% in children younger than five years of age. Molecules that are associated with hypersensitivity to eggs are mainly concentrated in egg white, with ovalbumin, lysozyme, ovomucoid, and ovotransferrin being the major egg allergens [9]. Some yolk-derived proteins have also been reported [9]. Egg allergy usually develops within the first five years of life, with 50% of children outgrowing egg hypersensitivity by three years [10,11]. Fortunately, in most cases, the prevalence of egg allergy decreases with age [12] and usually, it resolves by school age.

Eggs are of particular interest from a nutritional point of view, gathering essential lipids, proteins, vitamins, minerals, and trace elements [13], while offering a moderate calorie source (about 140 kcal/100 g), great culinary potential, and low economic cost. Indeed, eggs have been identified to represent the lowest-cost animal source for proteins, vitamin A, iron, vitamin B12, riboflavin, choline, and the second lowest-cost source for zinc and calcium [14]. In addition to providing well-balanced nutrients for infants and adults, egg contains a myriad of biologically active components [15,16,17]. These components are allocated in the various internal egg components (Figure 1). It has to be mentioned that eggshell and its tightly associated eggshell membranes are usually not consumed, although eggshell membranes are edible (Figure 1). The average consumption of eggs/year/capita in the world ranges from 62 (India) to more than 358 (Mexico) [18] and is even less in African Countries (36 eggs/year/capita) ([19]. Table eggs that are commercialized are not fertilized and are produced by about 3 billion hens, specifically bred throughout the world for human consumption.

Egg components are also reported to be highly digestible although a small amount of egg proteins is not assimilated [20], especially when egg is consumed as a raw ingredient [20,21,22]. The higher digestibility of cooked egg proteins results from structural protein denaturation induced by heating, thereby facilitating hydrolytic action of digestive enzymes. However, although the assimilation of egg protein is facilitated by heat-pretreatment and at a high level (91–94% for cooked egg-white proteins), it remains partly incomplete. It is noteworthy that major proteins, essentially egg-white proteins such as the proteinase inhibitor ovomucoid, and major egg-white ovalbumin resist thermal heating [23,24]. This observation is particularly interesting, knowing that egg-derived proteins and many hydrolytic peptides generated in vitro from limited digestion of egg-white proteins possess biological activities of interest for human health and may thus be used as nutraceuticals [16]. Indeed, several of those have been shown to exhibit antimicrobial, antioxidant, and anti-cancerous properties [25,26,27]. Thus, many authors have highlighted the importance of protein-derived peptides in the gut and their substantial role in the body’s first line of immunological defense, immune-regulation, and normal body functioning [28].

## 2. Egg Nutrients

Egg proteins are distributed equally between egg white and egg yolk, while lipids, vitamins, and minerals are essentially concentrated in egg yolk (Figure 2). Water constitutes the major part of egg (Figure 2) and it is noteworthy that the egg is devoid of fibers. The relative content of egg minerals, vitamins, or specific fatty acids may vary from one national reference to another [29] but remains globally comparable when considering major constituents such as water, proteins, lipids, and carbohydrates. The major egg nutrients are, indeed, very stable and depend on the ratio of egg white to yolk in contrast to minor components, which are affected by several factors including hen nutrition (See Section 4.2). In a whole, raw, and freshly laid egg, water, protein, fat, carbohydrates, and ash represent about 76.1%, 12.6%, 9.5%, 0.7%, and 1.1%, respectively [30].

### 2.1. Macronutrients

#### 2.1.1. Proteins

Egg white and egg yolk are highly concentrated in proteins. Hundreds of different proteins have been identified and are associated with specific physiological functions to fulfill time-specific requirements during embryo development. The compartment-specificity of some of these proteins can be explained by the fact that egg yolk and egg white are formed by distinct tissues. Egg yolk has essentially a hepatic origin, while egg white is synthesized and secreted after ovulation of the mature yolk in the hen’s oviduct [31].

The concentration of proteins is, on average, 12.5 g per 100 g of whole raw fresh egg, while egg yolk with its vitelline membrane and egg white contain 15.9 g protein and 10.90 g protein per 100 g, respectively. These values are slightly modified by hen genetics and age (See Section 4). Thanks to complementary proteomic approaches, nearly 1000 different proteins have been identified in the chicken egg, including the eggshell [32,33,34,35,36,37,38,39,40].

Yolk is a complex milieu containing 68% low-density lipoproteins (LDL), 16% high-density lipoproteins (HDLs), 10% livetins and other soluble proteins, and 4% phosvitins. These components are distributed between non-soluble protein aggregates called granules (19–23% of dry matter), which account for about 50% of yolk proteins, and a clear yellow fluid or plasma, that corresponds to 77–81% of dry matter [41,42]. Apolipoprotein B, apovitellenin-1, vitellogenins, serum albumin, immunoglobulins, ovalbumin, and ovotransferrin are the most abundant proteins of egg yolk, representing more than 80% of total egg-yolk proteins [43]. Yolk is tightly associated with the vitelline membranes, which consist of two distinct layers [44] that form an extracellular protein matrix embracing the yolk. These membranes provide to the yolk a physical separation from other egg compartments and prevents subsequent leakage of egg yolk towards egg white.

The egg white is a gel-like structure that lacks lipids and is composed mainly of water (about 88%) [44] (Figure 2), fibrous structural proteins (ovomucins), glycoproteins (ovalbumin, protease inhibitors), antibacterial proteins (lysozyme), and peptides (see Section 3.1) [33,45]. The average volume of egg white is estimated to be 30 mL (for an egg weighting 60 g, eggshell included) and protein concentration is about 110 mg/mL of egg white. In total, 150 distinct proteins have been identified in egg white [35], knowing that the very abundant ovalbumin accounts for 50% of the total egg-white proteins. The physiological function of this protein in egg remains unknown but ovalbumin is assumed to provide essential amino-acids for the chicken embryo growth. Egg-white ovalbumin thus represents a valuable source of amino-acids for human nutrition. Besides ovalbumin, egg white is concentrated in antibacterial lysozyme that is currently used as an anti-infectious agent in many pharmaceuticals and as food preservative (see Section 3.1). The viscous aspect of egg white is essentially due to ovomucin [46]. Remarkably, egg white is also characterized by the presence of four highly abundant protease inhibitors [47] that may delay digestion of egg components, especially when egg white is used as a raw ingredient in some food preparations.

#### 2.1.2. Lipids

The total lipid content is relatively stable in the egg ranging from 8.7 to 11.2 per 100 g of whole egg, when considering various EU countries and USA egg composition tables [29]. These lipids are only concentrated in the egg yolk (Figure 2 and Table 1) and a small part may remain tightly associated with vitelline membranes [48,49].

Lipids are part of yolk lipoproteins whose structure consists of a core of triglycerides and cholesteryl esters, surrounded by a monolayer of phospholipids and cholesterol in which apoproteins are embedded [42]. It is very difficult to change total lipid content in egg. An increase of fat in egg essentially depends on the increase of yolk-to-egg-white ratio, which is, however, poorly affected by hen’s nutrition. In contrast, fatty acid profile strongly depends on hen’s diet (See Section 4.2). This variability is illustrated in Table 1 by minimal and maximal values of fatty acids (saturated, monounsaturated, and polyunsaturated). Noticeably, the relative amount of unsaturated (monounsaturated + polyunsaturated) to saturated fatty acids in yolk (5.31 g versus 2.64 g per 100g of whole egg, Table 1) is particularly high compared to other animal-derived food sources. Yolk is also a rich source of essential fatty acids such as linoleic acid (FA 18:2 9c,12c (n-6)). The high content of cholesterol in eggs (400 mg per 100 g of whole egg) has contributed to the decline of egg intake 30 to 40 years ago, although many studies conducted in the 1990s have reported an absence of correlation between egg intake and high level of plasmatic cholesterol [3,4]. It is now assumed that variation in plasmatic cholesterol and associated cardiovascular disease risk results from food factors but also saturated fatty acids intake (such as dietary myristic (14:0) and palmitic (16:0) acids). Old studies performed in vivo on monkeys and gerbils have shown that dietary 14:0 (myristic acid) and 18:2 (linoleic acid) are the main fatty acids modulating plasma cholesterol—14:0 was the main saturated fatty acid raising plasma cholesterol and 18:2 was the only fatty acid that consistently lowered it [50,51]. In egg, 14:0 (myristic acid, 0.024 g per 100 g whole egg) is much less concentrated compared to unsaturated fatty acids 18:2 (linoleic acid, 1.38 g per 100g whole egg). These data all concur to corroborate that egg is not associated with higher cardiovascular disease incidence in healthy patients. However, egg intake has to be controlled in hyper-responders to dietary cholesterol (about 15% to 25% of the population), as an increase in egg consumption in these people affects plasma lipids to a greater extent than in hypo-responders.

#### 2.1.3. Carbohydrates

Egg does not contain any fibers and its content in carbohydrates is low (0.7%). Egg carbohydrates are distributed between egg yolk and egg white (Figure 2). Glucose is the dominant free sugar in the egg (about 0.37 g per 100 g of whole egg) and is essentially present in egg white (0.34 g per 100 g of egg white versus 0.18 g per 100 g of egg yolk) [30]. Trace amounts of fructose, lactose, maltose, and galactose have been detected in raw egg white and raw egg yolk [30]. Carbohydrates are also highly represented in egg proteins, knowing that many of them are glycoproteins undergoing post-translational glycosylations prior to secretion by reproductive tissues of the hen to form yolk, membranes, and egg white.

### 2.2. Micronutrients

#### 2.2.1. Vitamins and Choline

The egg and, more precisely, the egg yolk, is a vitamin-rich food that contains all vitamins except vitamin C (ascorbic acid). The absence of vitamin C in the egg may result from the fact that birds are capable of satisfying their own vitamin C requirements, by de novo synthesis from glucose [52]. The ability to produce vitamin C has been lost during the process of evolution in several animal species including guinea pigs, monkeys, flying mammals, humans, and some evolved passerine birds [52]. Consequently, these latter species, but not domestic birds, are dependent on dietary sources of vitamin C (fruits and vegetables). The egg yolk contains high amount of vitamin A, D, E, K, B1, B2, B5, B6, B9, and B12, while egg white possesses high amounts of vitamins B2, B3, and B5 but also significant amounts of vitamins B1, B6, B8, B9, and B12 (Table 2). Eating two eggs per day covers 10% to 30% of the vitamin requirements for humans. It is noteworthy that the content of liposoluble vitamins (vitamins A, D, E, K) in egg yolk is highly dependent on the hen’s diet (See Section 4.2). In addition to these vitamins, eggs represent a major source of choline, which is essentially concentrated in the yolk (680 mg/100 g in the egg yolk versus 1 mg/100 g in the egg white) [30,53]. It has been reported that hard-boiled egg represents the second major source of choline after beef liver [54] and the first source of choline in the US diet [55]. In foods, choline is found as both water-soluble (free choline, phosphocholine, and glycerophosphocholine) and lipid-soluble forms (phosphatidylcholine and sphingomyelin) and has important and diverse functions in both cellular maintenance and growth across all life stages. It plays some roles in neurotransmission, brain development, and bone integrity [54,56,57].

#### 2.2.2. Minerals and Trace Elements

Egg is rich in phosphorus, calcium, potassium, and contains moderate amounts of sodium (142 mg per 100 g of whole egg) (Table 3). It also contains all essential trace elements including copper, iron, magnesium, manganese, selenium, and zinc (Table 3), with egg yolk being the major contributor to iron and zinc supply. The presence of such minerals and micronutrients in egg is quite interesting as a deficiency in some of these (Zn, Mg, and Se) has been associated with depression and fatigue [59] and development of pathological diseases. The concentration of some of those trace elements (selenium, iodine) may be significantly increased depending on hen’s diet (See Section 4.2).

### 2.3. Antinutritional Factors

As mentioned above, major proteins of egg include protease inhibitors that may delay the proper degradation of egg proteins by inhibiting digestive enzymes including pepsin, trypsin, and chymotrypsin. Indeed, egg white is a major source of ovostatin, ovomucoid, ovoinhibitor, and cystatin [47]. Moreover, some of these molecules (ovoinhibitor, ovomucoid, cystatin) possess many disulfide bonds that are likely to confer moderate resistance to denaturation by proteases and gastric juices. Some of these antinutritional factors may be partly denatured by heat [20,22,23,24] during the process of cooking, thus facilitating protein access to digestive proteases. Moreover, some vitamin-binding proteins highly concentrated in egg may also limit some vitamin access—avidin that binds vitamin B12 (biotin) exhibits the highest known affinity in nature between a ligand and a protein [60]. The bioavailability of biotin for consumers may be compromised by the tight complex formed between avidin and its bound vitamin B8.

## 3. Egg Nutraceuticals

There is increasing evidence that egg is not solely a basic food of high nutritional value but that it also contains many bioactive compounds (lipids, vitamins, proteins, and derived hydrolytic peptides) [16,61,62,63,64] of major interest for human health. In vitro analyses performed on purified proteins have revealed a great potential in egg proteins as they exhibit a diversity of biological activities. Various tools combining physicochemical, analytical, and in silico approaches [65,66] can be used to identify hydrolytic peptides with potential bioactivities. It is remarkable that a lot of egg proteins have no identified physiological function described yet, besides providing essential amino-acids for the embryo but also for egg-eating species including humans. In addition to egg proteins displaying a wide spectrum of antimicrobial activities that could contribute to intestine health, many efforts have been made in the last decades to further characterize biological activities of egg-derived hydrolytic peptides that may naturally occur during the digestive process [20,22]. Interestingly, some of these bioactive peptides are specifically generated after limited proteolysis of denatured egg proteins [67], after boiling. Most of these studies were performed in vitro, but this finding opens many fields of research. To date, little is known on how egg proteins resist acidic pH of the stomach, digestive proteases, and intestinal microbiote, and how the presence of egg protease inhibitors in the diet can interfere with the degradation of egg proteins by digestive proteases. The kinetics of protein digestion is sequential, starting with the hydrolysis of proteins into peptides until complete degradation into dipeptides and, finally, free amino acids. But it is known that some egg proteins (ovalbumin, ovomucoid) are only partly digested [20,22] suggesting that some bioactive peptides may be generated naturally without undergoing complete degradation into amino acids.

### 3.1. Antimicrobials

Egg antimicrobials in edible parts are essentially concentrated in egg white and the vitelline membrane. Depending on the protein considered, these antimicrobials may exhibit antibacterial, antiviral, antifungal, or antiparasitic activities (Table 4).

Their antibacterial effect relies on several bactericidal or bacteriostatic mechanisms. Some of them have a powerful activity via interaction with bacterial walls that further triggers permeabilization and bacterial death (lysozyme, avian beta-defensins, etc.). The effects of the other molecules are rather indirect by decreasing the bioavailability of iron (ovotransferrin) and vitamins (avidin) that are required for some microbial growth, and by inhibiting microbial proteases that are virulent factors of infection (ovoinhibitor, cystatin) [68]. The various egg antimicrobial molecules that have been described so far in literature are listed in Table 4. Interestingly, some of them (AvBD11, OVAX, avidin, beta-microseminoprotein) are not expressed in the human genome [69], suggesting that they may constitute powerful anti-infectious agents against human enteric pathogens, to reinforce intestinal host immunity.

In addition to these egg proteins and peptides, there are increasing data reporting the antimicrobial activity of egg-derived peptides that may be released after partial hydrolysis by exogenous proteases. Such hydrolytic peptides obtained from lysozyme [70,71,72,73], from ovotransferrin [25], from ovomucin [74], and from cystatin [75] have shown a broad range of antibacterial activities.

### 3.2. Antioxidant Activities

Long-term oxidative stress in the gastrointestinal tract can lead to chronic intestinal disorders and there is increasing interest in investigating the potential of food-derived antioxidants, including egg antioxidants, in intestinal health. Chicken egg contains many antioxidant compounds that encompass vitamins, carotenoids, minerals, and trace elements but also major egg-white proteins [103,104,105,106] such as ovotransferrin, in its native form or as hydrolytic peptides [98,99,104,105,107,108,109,110], ovomucoid and ovomucoid hydrolysates [111,112], ovomucin hydrolysates and derived peptides [112], and egg yolk-proteins including phosvitin [113]. Most of these molecules have been generated in vitro but some assays performed in a porcine model have revealed the beneficial effect of proteins derived from egg yolk in reducing the production of pro-inflammatory cytokines [114]. The authors concluded that supplementation of the diet with egg yolk-proteins may be a novel strategy to reduce intestinal oxidative stress [114].

### 3.3. Anti-Cancerous Molecules

There are only few data showing that food-derived proteins and peptides can also be beneficial to prevent and to cure cancer diseases [26]. Several studies have confirmed the tumor-inhibitory activity of egg white lysozyme using experimental tumors. Its effect essentially relies on immunopotentiation [115]. Ovomucin (beta subunit) and ovomucin-derived peptides also showed anti-tumor activities via cytotoxic effects and activation of the immune system [74]. The anticancerous effect of egg tripeptides [27] and hydrolytic peptides from ovotransferrin [116] have also been published. Information in this field is quite scarce, but it may be worth continuing to investigate such activities. Some interesting data may arise from studies on egg protease inhibitors [47] since similar molecules existing in other food product, including legumes like pea, have been described as potential colorectal chemopreventive agents [117].

### 3.4. Immunomodulatory Activities

Several egg proteins have potential immunomodulatory activities. Among these, egg-white lysozyme is a promising agent for the treatment of inflammatory bowel disease. In a colitis porcine model, lysozyme was demonstrated to significantly protect animals from colitis and reduce the local expression of pro-inflammatory cytokines while increasing the expression of the anti-inflammatory mediators [118]. Sulfated glycopeptides generated by proteolysis from ovomucin, chalazae, and yolk membrane can exhibit macrophage-stimulating activities in vitro [74]. Cytokines, such as egg-white pleiotrophin, play a pivotal role in the generation and resolution of inflammatory responses. In human, pleiotrophin have been shown to promote lymphocyte survival, and to drive immune cell chemotaxis [119,120]. But, the biological significance of the potential immunomodulatory activity of egg white pleiotrophin in human intestine remains very speculative. In contrast, some valuable immunomodulatory activities might emerge from ovotransferrin and egg yolk vitellogenin hydrolysates [121,122] after partial degradation by digestive proteases.

### 3.5. Antihypertensive Activities

Considering the prevalence and importance of hypertension worldwide (over 1.2 billion individuals) [123], there is increasing ongoing research to find ways to regulate this multifactorial disease. At the population level, the most important factors of long-term control of blood pressure are sodium and potassium intakes and the importance of the renin-angiotensin-aldosterone system. Most egg-derived peptides with anti-hypertensive activities exhibit inhibitory activities against the angiotensin-converting enzyme (ACE). This enzyme triggers the processing and activation of angiotensin I into the active vasoconstrictor angiotensin II. Several yolk-derived peptides bearing antihypertensive activities have been described in the literature [113,124] along with ovotransferrin and egg white hydrolysates [125,126]. Some of these peptides contain only three amino-acids [27,127]. Some of these tripeptides were demonstrated to be active in vivo—the oral administration of these peptides that have been administrated orally in hypertensive rats contributed to significantly reduce blood pressure [128] and thus, may help in diminishing the occurrence of cardiovascular diseases [127,129].

## 4. Factors Affecting Egg Quality

### 4.1. Genetics

Selection for egg quality is an important component of the breeding strategies of companies that market egg-laying type hens. Indeed, consumers demand high quality products with strong eggshell, while reducing cost, guaranteeing eggs devoided of microbial contaminants, and improving the acceptability of rearing systems [130,131]. Most selection strategies to improve egg quality has focused on the shell’s physical properties (and its ability to resist shocks), stability of egg weight, egg-white quality, and yolk percentage. Egg-white quality essentially refers to albumen height that reflects freshness, and its ability to prevent microbial growth and survival that may be associated with toxi-infection risks for consumers (Salmonellosis). Recently, some authors reported differences in albumen height/pH in various selected lines [132] and corroborated that selection on specific traits has changed the proportion of the yolk, albumen, and shell while increasing albumen height. The moderately high variability in weights of egg yolk and albumen was also observed when comparing selected and traditional lines [133]. Egg white is a very unfavorable medium for bacteria, due to its high viscosity, its pH becoming progressively alkaline upon egg storage (7.8 to 9.5), and to the presence of a myriad of antimicrobial molecules (See Section 3.1). In fact, the antibacterial potential of egg white was demonstrated to be moderately inheritable [134]. Concerning egg proteins and peptides, some difference in relative abundance of certain molecules have been reported in brown versus white eggs or in different lines, but the major egg proteins remain basically unchanged [135,136].

### 4.2. Nutrition and Rearing Systems

Nutrition of laying hens, the characteristics of the feed (composition in nutrients, energy content, but also feedstuff texture and presentation), and the mode of feed delivery throughout the day affect, not only egg weight, but also to a lesser extent, egg-yolk and egg-white proportion [137]. The quality of the diet for pullets will influence the egg weight essentially at the onset of lay but it is much less significant when considering the whole period of lay [137]. The dietary characteristics include the level of calcium supply as well as its particular size. Dietary calcium in a particular form allow hens to express a specific appetite for calcium at the end of the day, that is stored and further assimilated during the night when the shell formation take place [138]. The egg weight is influenced by the daily energy consumption of hens. High energetic diet and the dietary supply of linoleic acid increase the egg weight. This effect is particularly relevant at the onset of laying (22–32 weeks) and much less pronounced in older hens [139]. Egg weight is also increased by the level of dietary proteins, and some studies have revealed that methionine was the main limiting amino acid as its presence in the hen’s diet is positively correlated with egg weight [140]. Additionally, energy intake depends on the protein source considering that laying hens are traditionally fed corn, wheat, and soybean meal. The presence of antinutritional factors in the diet (protease inhibitors and proteins that are highly resistant to digestive proteases such as convicilin, glycinin, cruciferin [141]) may affect the overall digestibility of the feed by the hens and subsequent egg weight. However, the content in major egg components is relatively stable and the variability depends essentially on the proportion of the albumen to the yolk, which exhibit very contrasted composition (See Section 2). In contrast, the fatty-acid profile of the yolk and the content of micronutrients such as vitamins and trace minerals (See Section 2.2) or of carotenoids are very variable and directly rely on diet composition [137].

The fatty-acid profile of an egg, incorporated in triglycerides and phospholipids, directly reflects the composition of fatty acids in the hen’s diet. In contrast, the enrichment of the hen’s diet with saturated fatty acids has less influence of the yolk lipid profile. The saturated and unsaturated fatty-acid content of the hen’s diet can be modified by the inclusion of oil or feedstuffs showing a high degree of unsaturated fatty acid such as fish, chia, flaxseeds [142,143], olive, or soy oils (see Reference [144] for a review). For instance, including olive oil in a hen’s diet favors the incorporation of mono-unsaturated fatty acids (in particular, oleic acid content) in the yolk, while soil oil increases that of unsaturated n-6 fatty acids (linoleic acid) [144]. More recently, results obtained with an enrichment of the diet with microalgae or linseeds have revealed the potential of these compounds to increase the content of yolk in n-3 fatty acids (more than 3- and 4-fold increase, respectively) [145]. A similar trend in increasing the polyunsaturated content of yolk was observed with marigold extract powder [146], Schizochytrium microalgae [147], prebiotic and probiotic combination [148], etc. To conclude, it is relatively easy to enrich the egg in some unsaturated fatty acids, of interest for human health. The challenge remains to identify animal and vegetable sources of polyunsaturated fatty acids that increase the content of these fatty acids in yolk without affecting its technological and/or sensorial quality, to suit both the food industry requirements (egg products) and consumer demand.

The content of some trace minerals in eggs such as selenium, iodine, and, at lower magnitudes, iron, zinc, fluoride, or magnesium can also be increased by larger dietary supply for hens [149]. The average selenium egg content is around 5 µg per egg and can be increased 3- to 6-fold (12-fold in the albumen and 4-fold in the yolk) and reach 30–40 µg/egg when hens are supplied with 0.3 to 0.5 mg selenium (from selenomethionine or selenium enriched yeast/kg diet). Such egg enrichment provides 50–70% of the daily human requirement [150].

Similarly, hen nutrition is a way to enrich egg in lipophilic vitamins (A, D, E, K) or water-soluble vitamins (folate, B12, pantothenic acid, and, at lower magnitudes, riboflavin, thiamin, and biotin). Egg content in vitamin A can be enhanced 10-fold from its initial value when hens are supplied with 30,000 IU retinol and that of vitamin D3 by 15-fold (2–5 to 34 µg/100g in hens fed 2500 and 15,000 IU D3). Vitamin E content in the yolk can increase 3- to 20-fold depending on basal diet content and dietary supply. For water-soluble vitamins, the magnitude of increase by larger dietary supply is lower—more than 2-fold for folate, riboflavin, or cobalamin and, at a lower level, thiamin, biotin and panthotenic acid, pyridoxine, or niacin [137]. Yolk color (yellow/orange shading) is also determined by the content in carotenoids in the diet [137]. The main sources of carotenoids (lutein, xanthophylls, and zeaxanthin) for birds are corn, lucerne, flower (marigold), and paprika extract (red carotenoids) that are incorporated in a hen’s diet to meet consumer demand for a more orange-yellow yolk. Beside its interest in increasing the visual aspect of the yolk, the high content of carotenoids in the yolk may also have a positive incidence for human health in increasing visual performance and reducing the risk of age-related macular degeneration [151].

Because birds in free-rearing systems have access to grass, insects, and worms in addition to their basic diet, the content in some egg micronutrients may also slightly vary. As an example, free rearing results in significantly higher total tocopherol, alpha-tocopherol, and lutein contents, as compared to the battery cage and the organic system, respectively, when hens are fed a similar conventional diet [152]. Conversely, no significant differences were observed in the lipid and total sterol contents [152]. A decrease in egg-white height and in yolk color was also observed when comparing egg from conventional cages to free-range systems [153]. However, eggs from conventional systems contain, in general, more carotenoids and vitamins due to the possibility of including chemical additives in the diet, knowing that such practice is not conducted in organic systems. In parallel, since the immune system of the hens is likely to be more challenged by the presence of environmental microbes in free-range systems, an increase of immunoglobulin Y content in egg yolk (initially to provide some passive immunity for the chick, similarly to maternal colostrum for infants) is also likely to increase. Furthermore, some have shown that the antimicrobial capacity of egg white may also be slightly modulated when hens are exposed to environmental microbes [154]. To conclude, rearing laying hens in free-range systems may globally improve the antimicrobial potential of eggs (See Section 3.1).

### 4.3. Physiological Status

Egg production and quality are greatly influenced by the physiological status of the hen—age, stress, and immune status [155]. Egg weight varies from 50 g to 70 g depending mainly on hen’s age and genetics (See Section 4.1). In modern flock, the egg weight has been restricted to 62–66 g throughout the laying cycle (20 to 80 weeks of age). Egg weight is the primary criterion used in egg grading (small, medium, large, extra-large). The increase of egg weight observed in older hens is associated with an increase in albumen and yolk average weights and in relative yolk proportion [156]. Hen age is also associated with a decrease in eggshell strength, in albumen height (the higher albumen height, the higher the freshness grade) [153], and a decrease in the strength of vitelline membrane that is frequently associated with a higher incidence of yolk rupture [157]. This latter observation is likely to result from an increase in yolk proportion in older hens. However, when considering the chemical composition of eggs, the results are quite controversial, although some have shown some difference in fatty-acid composition (docosahexaenoic acid and arachidonic acid) of egg depending on hen’s age [158]. Overall, these alterations in egg quality with age are, however, consistent with physiological changes and maybe some primary metabolic dysfunctions occurring in commercial hens at the end of the laying cycle [159].

Egg-laying performance also depends on general hen’s health as diseases and infections can induce loss of appetite and physiological failure thereby affecting animal growth, egg laying, and egg quality (eggshell deformation, eggshell defects, egg-white thinning, etc.). The most commonly found microbes in laying hens that affect hygienic egg quality are Salmonella enterica Enteritidis, mycoplasma, infectious bronchitis virus, and avian influenza virus [160]. Another major issue for hen welfare and egg quality is the poultry red mite that is found in most rearing systems for laying hens, regardless of the system type [161]. As a blood-feeder, this mite has dramatic effects on fowl host welfare including distress, anemia, reduced egg production, and reduced egg quality [162]. The prevalence of red mites is expected to increase, as a result of recent hen husbandry legislation changes (in favor of non-cage systems), increased acaricide resistance, climate warming, and the absence of efficient and sustainable solutions to control infestations [163]. It is also a major concern for public health as it may be a vector for food-borne pathogens including Salmonella species [164]. Amongst all these microbes, Salmonella enterica Enteritidis is the most critical for egg consumers as this pathogen can survive in egg white [165], even after several weeks of storage at 4 °C and 25 °C [166] and be responsible for food-borne diseases. It remains the dominant pathogen associated with egg consumption [167]; but, considerable efforts have been made by authorities to control Salmonella Enteritidis dissemination throughout the egg production chain. Reported salmonellosis case numbers continued to decrease thanks to the implementation of successful Salmonella control programs in poultry production [168], including Salmonella detection and monitoring [169], the establishment of pre-harvest measures [170], management and sanitation measures [171], and egg decontamination by washing under certain conditions [172]. It is noteworthy that European legislation does not allow egg washing in Europe (Commission Regulation (EC) No 589/2008), “because of the potential damage to the physical barriers, such as the cuticle, may favor trans-shell contamination with bacteria and moisture loss and thereby increase the risk to consumers, particularly if subsequent drying and storage conditions are not optimal”.

### 4.4. Egg Storage and Heat Treatment

Shell eggs are stored at room temperature or preferably in the fridge prior to be used by consumers (eggs are considered as “fresh” up to 28 days after laying). The conditions of egg storage can induce deep internal changes including physicochemical modifications that may increase some technological properties that are useful for the food industry, and alteration of antibacterial properties of the egg white [173,174,175,176] (Figure 3). These alterations result from water exchange between yolk and the egg white and from water and carbon dioxide loss through the eggshell pores, which elicited an increase in the air cell that develops between the two eggshell membranes (Figure 3a). Albumen height decreases with time of storage while albumen pH and whipping volume increase [136]. In parallel, the strength of vitelline membrane decreases upon egg storage due to loosening thereby impacting yolk shape/index (the yolk becomes flat and its diameter is higher) [173]. These latter modifications favor egg white/egg yolk exchanges of components such as carbohydrates and glucose [177], proteins [178,179,180], vitamins, and trace elements [181]. In addition, storage duration and conditions are associated with protein degradation [175,179,180] and a decrease in its antibacterial potential [175]. However, except for proteins, there is only little information available that describes the changes/denaturation in lipids, vitamins, and minerals composing both egg white and egg yolk during storage. It will be interesting to further investigate how these modifications impact the respective functional, nutritional, and technological properties of egg yolk and egg white (foaming, emulsifying properties, etc.). Recent data has demonstrated that the antioxidant activity of egg yolk was globally unchanged during six weeks of retail storage [182]. All these alterations of freshness criteria are accelerated at room temperature when compared to refrigerated conditions.

In addition to storage, one would expect that egg nutrients may also be modified during cooking. No clear evidence of minerals or vitamins denaturation could be observed when comparing fresh, soft-boiled, and hard-boiled egg (Table 5).

Actually, some data appear contradictory from one reference source to another (CIQUAL versus USDA, Table 5). Anyhow, it seems that the amount of polyunsaturated fatty acids, selenium, and vitamin A [21] tends to decrease upon cooking, especially in hard-boiled eggs (Table 5).

Noticeably, proteins undergo major conformational modifications upon cooking, even though their relative amount is not impacted by cooking (Table 5). This protein denaturation may be beneficial to inactivate antinutritional factors such as egg-white antiproteases but also to denature highly resistant proteins, thereby facilitating protease activity in the digestive tract. A higher digestibility of egg proteins may also contribute to limiting hypersensitivity to eggs in children [183,184]. Meanwhile, it has been shown that cooking significantly reduces the oxygen radical scavenging capacity (antioxidant potential) of egg yolk associated with free aromatic amino acids, lutein, and zeaxanthin [182], and also impacts lipids of yolk [185]. These observations corroborate that taking into account the food matrix and the way eggs are prepared is of major importance to appreciate egg digestibility and its associated nutritional and nutraceutical quality [186]. To conclude, it is fairly difficult to assess the beneficial/risk balance of cooking egg for human health as many molecules may be affected by cooking, while, in parallel, the heating process may increase the digestibility of egg proteins and potentially reveal potential new bioactive peptides [187,188]; but, it is worth mentioning that cooking eggs also allows for the elimination potential pathogens responsible for toxi-infections in consumers. In conclusion, taking into account all these data, the advice to retain most nutritional and nutraceutical benefits associated with egg would be to foster the consumption of poached or soft-boiled eggs, where the egg white is cooked (to inactivate antinutritional factors and potential pathogenic bacteria) while the egg yolk remains essentially raw (to preserve most vitamins, lipids, micronutrients, and some bioactive (antioxydant) molecules).

### 4.5. Variability between Avian Domestic Species

The table-egg market is dominated by chicken eggs in all countries. However, duck eggs are also widely consumed in some Asian countries. The major reason for this ascendancy of chicken eggs relies on several reasons—chickens are easy to handle and to rear and they have been selected for decades to lay close to 320 eggs a year. Conversely, geese, turkeys, and ducks are seasonal layers and require most specific sanitary and rearing conditions. Chicken eggs are also of reasonable size, not too big and larger than quail eggs, the latter being eaten occasionally as a gourmet ingredient.

Although egg compositions from traditional domestic species share common characteristics [189,190], they possess some significant differences in terms of energy that are mostly explained by the change in relative proportion of yolk to egg white (Figure 4). The energy (kcal/100g) for chicken, quail, duck, goose, and turkey eggs is 143, 158, 185, 185, 171, respectively. While the relative amount of proteins remains stable between species (about 13%), the lipid proportion varies from 9.5% (chicken) to more than 13% (duck, goose) (Figure 4, which explains most of the variation in their respective energy value. Yolks from duck and goose species have relatively higher fat content and higher percentage of yolk compared to chicken egg [189]. In parallel, the lipid profile of egg yolk exhibits some specificities depending on species [191,192,193]. Overall, the composition of duck egg resembles that of goose egg, which is consistent with their phylogenetic proximity.

It is also noteworthy that the content of minerals and trace elements of chicken eggs are usually lower that those observed for the other species, particularly in duck and goose species (Table 6). A similar trend is observed for vitamins (Figure 5). However, it has to be mentioned that the change in vitamins and trace elements in eggs depends mainly on diet composition. Therefore, these differences might rather reflect conditions of bird rearing than genetic capacity of hens to retain these compounds in eggs.

Besides these chemical compounds, some variability in bioactive molecules were also reported. Indeed, the comparative analysis of egg-white and egg-yolk proteomes revealed some proteins that are specifically associated with one or the other species [194,195,196,197] and others (ovotransferrin, lysozyme, gallin), displaying difference in abundance [196]. Overall, these characteristics may affect the overall bioactive potential of egg-white extract (antibacterial and/or antioxidant activity) depending on its bird origin, and it was shown that egg white from chicken egg retains the highest antibacterial potential compared to egg white from turkey, duck, and goose, at least, against some bacterial strains (Bacillus subtilis, Staphylococcus aureus, Pseudomonas aeruginosa, and Escherichia coli) [91]. Furthermore, the lower abundance of ovomucoid (a protease inhibitor that is highly resistant to chemical and thermal denaturation) in egg white from duck, goose, or turkey [196] may be associated with higher protein digestibility.

To conclude, such differences in the chemical composition of egg, and also in some of its biologically active molecules, are likely to be correlated with an increased or lower nutraceutical value of some egg proteins from other species, compared with the chicken egg.

## 5. Conclusions

For ages, eggs have been considered as foods of high nutritional value for humans and are widely consumed worldwide. Its consumption is predicted to continuously increase in the future, considering the growing number of occidental consumers who start to adopt a meat-free diet (vegetarians) or who significantly reduce their meat intake. This change in our consumption mode and food habits is motivated by much data on the association risk of meat intake with digestive cancers and cardiovascular diseases, and growing numbers of studies that praise vegetarian diet [198,199,200,201]. In parallel, it is also driven by ethical concerns and environmental issues with regards to modes of meat production [202]. It has also to be highlighted the existence of substantial disparities in egg consumption between countries [18], which is particularly low in Central Africa, with only 36 egg/year/capita [19]. The development of the egg industry in developing countries may represent a great opportunity for human nutrition/health and economy.

Besides basic nutrients, eggs are also a great source of potential nutraceuticals. A total of 550 distinct proteins were identified so far in egg-white and yolk/vitelline membranes and the physiological function of only 20 of them is characterized to date. This remark suggests that the egg probably still encloses many unknown activities that merit further investigations considering the current lack of research assessing the fate of egg proteins along the digestive tract. Such studies could help to better appreciate the in vivo potential of egg proteins and of the resulting hydrolytic peptides, and could be easily apprehended using dynamic gastric models that have been used with other foodstuffs, in food- and pharmaceutically-based research [203,204]. These in vitro models mimic both the biochemical and mechanical aspects of gastric digestion. They incorporate artificial saliva, compressive forces to disintegrate food, simulate continuous gastric emptying, and gastric secretion that generate pH profiles similar to human stomach. They also include bile salts and intestinal enzymes that act sequentially in a realistic time-dependent manner [205] and may be improved by adding intestine-like microbiota. This model was already used in a wide range of studies in order to assess the bioaccessibility of nutrients and to study the structural changes of food matrices. It is expected that such an experimental strategy would constitute a promising way to study the impact of egg preparation in the diet (raw versus cooked) on the physiological generation of bioactive peptides, and to better appreciate their biological significance for human health. It is well assumed now that gut health depends on the interplay between host genome, nutrition, and lifestyle that it contributes to normal brain function and mental health [206].

## Figures and Tables

**Figure 1 nutrients-11-00684-f001:**
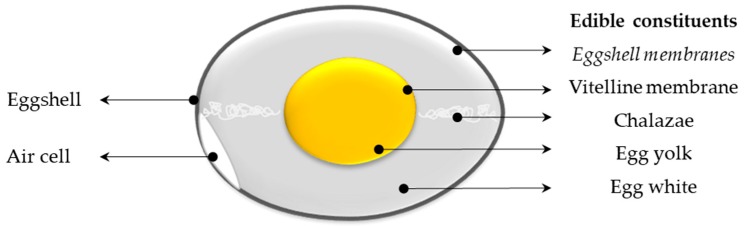
Egg structure. In italics: eggshell membranes are edible but usually not consumed, as they remain tightly associated with the eggshell.

**Figure 2 nutrients-11-00684-f002:**
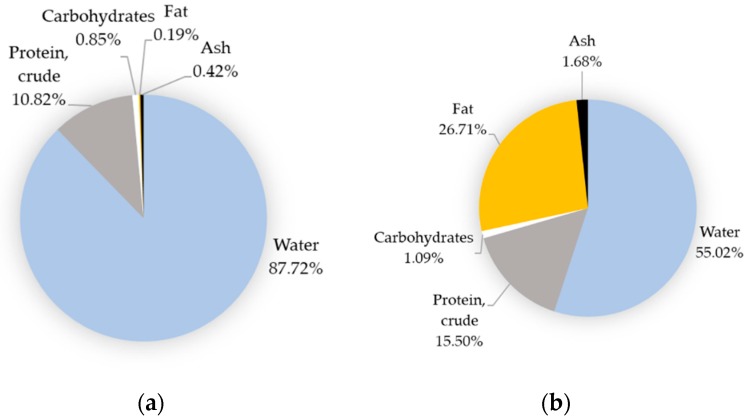
Basic composition of edible parts of the egg. (**a**) Egg white; (**b**) Egg yolk. Note that for (**b**), results refer to egg yolk/vitelline membrane complex. Retrieved on 01/11/2019 from the Ciqual homepage https://ciqual.anses.fr/ (French Agency for Food, Environmental and Occupational Health & Safety. ANSES-CIQUAL).

**Figure 3 nutrients-11-00684-f003:**
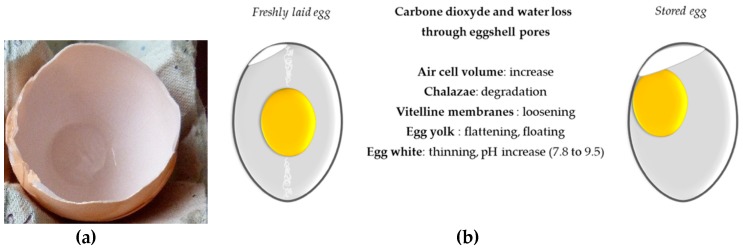
Physicochemical changes associated with egg storage (freshly laid egg versus egg stored for 2 weeks at room temperature). (**a**) air cell; (**b**) major modifications occurring during storage.

**Figure 4 nutrients-11-00684-f004:**
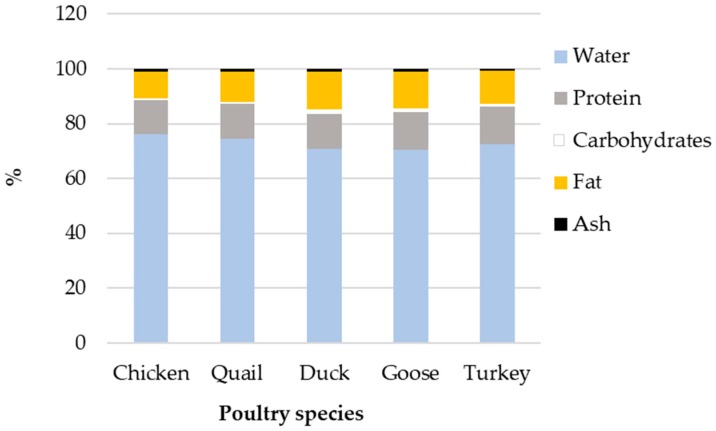
Variability of egg composition in five poultry species. Retrieved on 01/17/2019 from Department of Agriculture, Agricultural Research Service (2014). USDA National Nutrient Database for Standard Reference, Release 27. http://www.ars.usda.gov/ba/bhnrc/ndl/.

**Figure 5 nutrients-11-00684-f005:**
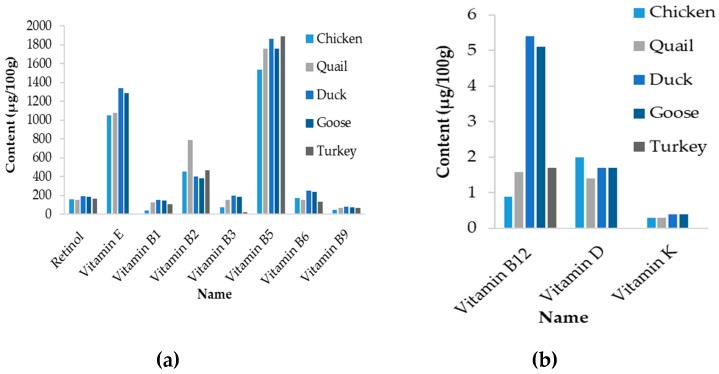
Vitamin profile of eggs from various domestic species. (**a**) Vitamins of high abundance; (**b**) vitamins of low abundance. Note that the concentration of vitamin E, D and K in turkey egg were not available. Retrieved on 01/17/2019 from Department of Agriculture, Agricultural Research Service (2014). USDA National Nutrient Database for Standard Reference, Release 27. http://www.ars.usda.gov/ba/bhnrc/ndl/.

**Table 1 nutrients-11-00684-t001:** Egg lipids ^1^.

Name	Egg, Whole, Raw			Egg Yolk, Raw		
	Average Content (g/100g)	Min. Value (g/100g)	Max. Value (g/100g)	Average Content (g/100g)	Min. Value (g/100g)	Max. Value (g/100g)
FA saturated	2.64	0.05	3.13	8.47	7.13	9.55
FA 4:0	<0.05	0	- ^2^	0	-	-
FA 6:0	<0.05	0	-	0	-	-
FA 8:0	<0.05	0	-	0.009	-	-
FA 10:0	<0.05	0	-	0.009	-	-
FA 12:0	<0.05	0	-	0.009	-	-
FA 14:0	0.024	0	0.038	0.091	0.077	0.1
FA 16:0	1.96	0.05	2.43	6.04	5.03	6.86
FA 18:0	0.65	0.05	0.89	1.73	-	2.42
FA monounsaturated	3.66	0.05	6.73	11.9	10.2	13.8
FA 18:1 n-9 cis	3.51	3.03	3.65	10.4	9.69	11.2
FA polyunsaturated	1.65	0.05	3.39	4.07	3.33	4.66
FA 18:2 9c,12c (n-6)	1.38	1.18	2.7	3.28	-	3.62
FA 18:3 9c,12c,15c (n-3)	0.061	0.02	0.58	0.15	-	0.27
FA 20:4 5c,8c,11c,14c (n-6)	0.12	-	0.13	0.37	-	0.4
FA 20:5 5c,8c,11c,14c,17c (n-3) EPA	0	-	0.003	0.01	-	0.011
FA 22:6 4c,7c,10c,13c,16c,19c (n-3) DHA	0.09	0.045	0.18	0.25	0.11	0.46
Cholesterol	0.398	0.344	0.423	0.939		1.280

^1^ Retrieved on 01/17/2019 from the Ciqual homepage https://ciqual.anses.fr/ (French Agency for Food, Environmental and Occupational Health & Safety. ANSES-CIQUAL). FA, fatty acids; EPA, eicosapentaenoic acid (omega-3 fatty acid); DHA, docosahexaenoic acid (omega-3 fatty acid). ^2^ not available.

**Table 2 nutrients-11-00684-t002:** Egg vitamins ^1^ (average content; µg/100g).

Name	Egg, Whole, Raw	Egg Yolk, Raw	Egg White, Raw
Vitamin A or Retinol	160	371	0
Vitamin A precursor or Beta-carotene	0	88	0
Vitamin D or Cholecalciferol	2.0	5.4	0
Vitamin E or Alpha-tocopherol	1050	2580	0
Vitamin K or Phylloquinone	0.3	0.7	0
Vitamin C	0	0	0
Vitamin B1 or Thiamin	40	176	4
Vitamin B2 or Riboflavin	457	528	439
Vitamin B3 or Niacin	75	24	105
Vitamin B5 or Pantothenic acid	1533	2990	190
Vitamin B6	170	350	5
Vitamin B8 or Biotin	16.5–53.8 ^2^	27.2–49.4 ^2^	5.7–7.9 ^2^
Vitamin B9 or Folate	47	146	4
Vitamin B12 or Colabamin	0.89	1.95	0.09

^1^ Retrieved on 01/17/2019 from Department of Agriculture, Agricultural Research Service (2014). USDA National Nutrient Database for Standard Reference, Release 27. http://www.ars.usda.gov/ba/bhnrc/ndl
^2^ [58].

**Table 3 nutrients-11-00684-t003:** Egg minerals and trace elements (average content; mg/100g) ^1^.

Name	Egg, Whole, Raw	Egg Yolk, Raw	Egg White, Raw
Calcium	56	129	7
Copper	0.072	0.077	0.023
Iodine	0.021	0.18	0.002
Iron	1.75	2.73	0.08
Magnesium	12	5	11
Manganese	0.028	0.055	0.011
Phosphorus	198	390	15
Potassium	138	109	163
Selenium	0.030	0.056	0.020
Sodium	142	48	166
Zinc	1.29	2.30	0.03

^1^ Retrieved on 01/17/2019 from Department of Agriculture, Agricultural Research Service (2014). USDA National Nutrient Database for Standard Reference, Release 27 http://www.ars.usda.gov/ba/bhnrc/ndl and from the Ciqual homepage https://ciqual.anses.fr/ (French Agency for Food, Environmental and Occupational Health & Safety. ANSES-CIQUAL) for iodine content.

**Table 4 nutrients-11-00684-t004:** Major egg antimicrobial proteins.

	Gene ID/Gene Symbol	Target Organisms	Localization ^1^	References
Avian beta-defensin 11	414876/AVBD11	Bacteria	EW, VM	[76,77]
Avidin	396260/AVD	Bacteria	EW, VM, EY	[78,79]
Beta-microseminoprotein-like	101750704	Bacteria	EW	[76]
Cystatin	396497/CST3	Bacteria, viruses, fungi, parasites	EW, VM, EY	[75,80,81,82,83,84,85]
Gallin	422030/OvoDA1	Bacteria	EW	[86,87,88,89]
Immunoglobulin Y	-		EY	[90]
Lysozyme	396218/LYZ	Bacteria, viruses, fungi	EW, VM, EY	[67,91,92]
Ovalbumin-related protein X	420898/OVALX	Bacteria	EW, VM, EY	[93]
OvoglobulinG2/TENP	395882/BPIFB2	Bacteria	EW, VM, EY	[94,95,96]
Ovoinhibitor	416235/SPIK5	Bacteria	EW, EY	[97]
Ovomucin (alpha and beta subunits)	395381/LOC395381 (alpha) 414878/MUC6 (beta)	Bacteria, viruses	EW, VM	[74]
Ovotransferrin	396241/TF	Bacteria, viruses	EW, VM, EY	[98,99,100]
Phosvitin	424547/VTG1	Bacteria	EY	[101]
Pleiotrophin	418125/PTN	Bacteria	EW	[76,102]
Vitelline membrane outer layer protein 1	418974/VMO1	Bacteria	EW, VM, EY	[76]

^1^ Edible parts: EW, egg white; EY, egg yolk; VM, vitelline membrane.

**Table 5 nutrients-11-00684-t005:** List of egg characteristics and major components that vary upon cooking ^1^.

Name	Egg, Whole, Raw, Fresh	Egg, Whole, Soft-Boiled	Egg, Whole, Hard-Boiled
Energy (kcal/100g)	140 ^1^; *143* ^2^	142; *143*	134; *155*
Protein (g/100g)	12.7; *12.56*	12.2; *12.51*	13.5; *12.58*
Carbohydrate (g/100g)	0.27; *0.72*	1.08; *0.71*	0.52; *1.12*
Fat (g/100g)	9.83; *9.51*	9.82; *9.47*	8.62; *10.61*
FA saturated (g/100g)	2.64; *3.126*	3.11; *3.11*	2.55; *3.267*
FA monounsaturated (g/100g)	3.66; *3.658*	4.42; *3.643*	3.57; *4.077*
FA polyunsaturated (g/100g)	1.65; *1.911*	1.28; *1.904*	1.03; *1.414*
Cholesterol (mg/100g)	398; *372*	222; *370*	355; *373*
Salt (g/100g)	0.31	0.2	0.31
Calcium (mg/100g)	76.8; *56*	150; *56*	41; *50*
Potassium (mg/100g)	134; *138*	164; *138*	120; *126*
Selenium (µg/100g)	30	23.8	7.01
Vitamin A, Retinol (µg/100g)	182; *160*	132; *160*	61.5; *149*
Vitamin D (µg/100g)	1.88; *2.0*	1.28; *2.0*	1.12; *2.2*
Vitamin E (mg/100g)	1.43; *1.05*	2.17; *1.04*	1.03; *1.03*
Choline (mg/100g)	250; *293.8*	-	230; *293.8*

^1^ Source: French Agency for Food, Environmental, and Occupational Health & Safety. ANSES-CIQUAL French food composition table version 2017. Retrieved on 01/11/2019 from the Ciqual homepage https://ciqual.anses.fr/; ^2^ In italics, retrieved on 01/18/2019 from Department of Agriculture, Agricultural Research Service (2014). USDA National Nutrient Database for Standard Reference, Release 27 and USDA Database for the Choline Content of Common Foods, Release 2 [30]; FA, fatty acid.

**Table 6 nutrients-11-00684-t006:** Comparison of egg minerals and trace elements in chicken, quail, duck, goose, and turkey eggs (average content; mg/100g).

Name	Chicken	Quail	Duck	Goose	Turkey
Calcium	56	64	64	60	99
Copper	0.072	0.062	0.062	0.062	0.062
Iron	1.75	3.65	3.85	3.64	4.1
Magnesium	12	13	17	16	13
Manganese	0.028	0.038	0.038	0.038	0.038
Phosphorus	198	226	220	208	170
Potassium	138	132	222	210	142
Selenium	0.0307	0.032	0.0364	0.0369	0.0343
Sodium	142	141	146	138	151
Zinc	1.29	1.47	1.41	1.33	1.58

Retrieved on 01/17/2019 from Department of Agriculture, Agricultural Research Service (2014). USDA National Nutrient Database for Standard Reference, Release 27. http://www.ars.usda.gov/ba/bhnrc/ndl.

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
