# Peer review of "The Golden Egg: Nutritional Value, Bioactivities, and Emerging Benefits for Human Health"

_nutrients, 2019, doi:10.3390/nu11030684_

Round 1
Reviewer 1 Report
A well done comprehensive overview but I miss more about the valuable tool as a source of DHA. Some references would be valuable for that in the paragraph lines 351-355. Sop m,uch is given for minerals and vitamins but the value to enrich with fatty acids is too short and should be extended.
Author Response
As required by the reviewer 1, the paragraph related to fatty acid profile has been developed and additional information has been included. The paragraph now contains 16 lines instead of 5, and 7 references instead of 1.
Reviewer 2 Report
The review entitled “The golden egg: nutrition value, bioactivities and emerging benefits for human Health” from Réhault-Godbert et al. has the aim to give both an overview of the main nutritional characteristics and a description of bioactive compounds and factors affecting quality of eggs. Authors provide a detailed and updated overview based on emerging data published. I recommend only few editing of text before the publication.
I suggest to modify the following points:Line 71: Change “Health” with “health”Line 99: Insert a space between “average” and “12.5”Line 552: Insert a space between “the” and “egg”
Table 4: In order to have a better form of the table, I propose the following changes: “Gene Name” with “Gene ID”; “Antimicrobial activity” with “Activity”; “Egg localization” with “Localization” and please check that every protein have a proper Gene ID/Gene Symbol. At last, I suggest the use of acronyms for the activities listed opportunely insert in legend.
Author Response
The suggestions of text editing have been taken into consideration and the text has been corrected accordingly. Moreover, the article has been read carefully to remove remaining grammatical errors. The reviewer suggested the use of acronyms for the activities listed in Table 4. We propose a new way to illustrate the activities by replacing antibacterial by bacteria etc. and by changing the title of the column (target organisms instead of antimicrobial activity). The accuracy of gene IDs and gene symbols has been verified.
Additional modification: Figure 5 has been resized for better reading.